# Barcoding Hymenoptera: 11 Malaise Traps in Three Thai Forests: The First 68 Trap Weeks and 15,338 Parasitoid Wasp Sequences

Donald L. J. Quicke [1] [iD], Paul D. N. Hebert [2,3] [iD], Mikko Pentinsaari [2] [iD] and Buntika A. Butcher [1,*]

1   Integrative Insect Ecology Research Unit, Department of Biology, Faculty of Science, Chulalongkorn University, Bangkok 10330, Thailand; d.quicke@email.com
2   Centre for Biodiversity Genomics, Guelph, ON N1G 2W1, Canada; phebert@uoguelph.ca (P.D.N.H.); mpentins@uoguelph.ca (M.P.)
3   Department of Integrative Biology, University of Guelph, Guelph, ON N1G 2W1, Canada
*   Correspondence: buntika.a@chula.ac.th

**Abstract:** We report the results of DNA barcoding week-long Malaise trap catches from 11 sites in three Thai conservation areas, concentrating on the parasitoid Hymenoptera, particularly the superfamily Ichneumonoidea. From a total of 15,338 parasitoid wasp sequences, 13,473 were barcode compliant and could be assigned to a family based on morphology and sequence data. These collectively represented 4917 unique BINs (putative species) in 46 families, with the Scelionidae, Ichneumonidae, Eulophidae, Braconidae and Platygastridae being, by far, the most abundant. Spatial proximity had a strong positive effect on the numbers of BINs shared between traps.

**Keywords:** BINS analysis; parasitic wasps; tropical fauna

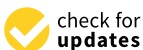



## 1. Introduction

It is well known that tropical forests have highly diverse insect communities. Just over 40 years ago, Erwin [1], when sampling beetles from the canopy of trees in Panama, carried out a "back of an envelope" calculation, extrapolating from the number of morphospecies he had found coupled with assumptions about global numbers of tree species, host specificity, and the proportions of insects with different life histories, and came up with the startling estimate that there might be as many as 30 million insect species in the world [2]. At that time, fewer than a million species had been described, and expert estimates tended toward the real total probably being around 2 million. In fact, one standard entomology textbook [3,4] estimated that there were only about 3 to 4 million extant insect species. Stork [5] reviewed the literature estimates and came to the conclusion that the high estimates were unreliable and that, based on what he considered robust studies, there were probably approximately 1.5 million species of beetles and 5.5 million species of insects in total, indicating that some 80% of species are still undescribed. However, it was pointed out that more work needed to be carried out on "less-studied taxa such as many families of Coleoptera, Diptera, and Hymenoptera" and, in particular, on less-well-studied faunas, essentially the tropics.

Novotny et al. [6] provided evidence that one of the assumed parameter values in the Erwin equation would lead to an overestimate, as their study showed low levels in terms of the host specificity of tropical herbivorous insects. In this paper, we consider the parasitoid wasps whose global diversity is also likely to be determined in part by levels of host specificity. Based almost entirely on evidence from the Holarctic, some generalisations can be made. Parasitoid hymenopterans collectively show a spectrum of biological attributes on whether their hosts continue development after the parasitisation event (the koinobiont strategy) or whether host development (feeding, growth) is stopped (the idiobiont strategy) [7–10].

Most studies, however, focus on the taxonomically better-known insect groups, such as Lepidoptera (especially butterflies) [11], ants, and beetles. The parasitoid Hymenoptera, despite being of enormous ecological importance, pose far greater identification challenges, and, if studied at all, this mostly involves using un-named morpho-species. Indeed, in many groups, the great majority of species are still undescribed.

For some years now, DNA barcoding has been used to help discriminate between cryptic species as part of taxonomic revisionary work [12,13]. When there is a far more complete barcode library for tropical insects, it may be possible to use DNA barcoding to provide species names for many individuals. However, species coverage in both GenBank and BOLD is still sparse when it comes to the megadiverse tropical faunas more than a decade after Kwong et al. noted this problem [14].

Even within parasitoids, there is considerable variation in developmental mode and host utilisation [8,9]. Here, we define parasitoid wasps as all those with a parasitoid lifestyle, including various aculeate wasp groups such as chrysidoids.

As regards parasitoid wasps, the most prolific study concerns the taxonomically difficult braconid subfamily Microgastrinae, which, based on focal taxon extrapolations, placed minimum and maximum bounds on the number of species in this one subfamily as 17,000 and 46,000+, respectively [15]. The higher estimate is almost the same as the total number of described members for the whole of Ichneumonoidea [16].

As one component of the Global Malaise Project, a total of 11 Malaise traps are being run for 12 months in three forested National Park and Conservation areas in north and central Thailand. Specimens collected using these traps are being barcoded, and, here, we present the data for all groups of Hymenoptera with parasitoid lifestyles with particular attention to the large superfamily Ichneumonoidea (comprising the families Braconidae and Ichneumonidae) because of their multiple independent biological and host transitions [10]. The data are explored concerning aspects of forest type, seasonality, parasitoid life history strategy, host group abundance/diversity, and species overlap using Barcode Index Number numbers (BINs) as a proxy for species [17].

## 2. Materials and Methods

### 2.1. Field Sampling

Standard ez-Malaise traps (Bugdorm, Taichung, Taiwan) (Figure S1A–K) were deployed in three conservation areas (Figure 1; Table 1): Khao Yai National Park and Sakaerat Environmental Research Station in Central Thailand, and Doi Phu Kha National Park in Northern Thailand. Detailed trap locations at each site are shown in Figure 2. The collection head sample bottles were filled with 96% ethanol, and the samples were collected every 7th day, held in a fridge, and shipped to the Centre for Biodiversity Genomics (CBG; University of Guelph, Guelph, Ontario, Canada) every three months for DNA barcode analysis.

**Table 1.** Details of Malaise trap locations, codes and habitat.

| Location | Trap Number | Trap Code | Forest Type | Latitude | Longitude | Elevation (m) |
|---|---|---|---|---|---|---|
| Khao Yai N.P. | 1 | THAMA | Dry evergreen forest | 14°28.256′ N | 101°22.396′ E | 779 |
| Khao Yai N.P. | 2 | THAMB | Dry evergreen forest | 14°26.800′ N | 101°22.017′ E | 752 |
| Khao Yai N.P. | 3 | THAMC | Secondary forest | 14°26.016′ N | 101°22.153′ E | 732 |
| Khao Yai N.P. | 4 | THAMD | Secondary forest | 14°25.548′ N | 101°23.078′ E | 697 |
| Sakaerat | 1 | THAME | Dry dipterocarp forest | 14°30.336′ N | 101°56.147′ E | 353 |
| Sakaerat | 2 | THAMF | Dry dipterocarp + dry evergreen forest ecotone | 14°30.580′ N | 101°55.980′ E | 365 |
| Sakaerat | 3 | THAMG | Dry evergreen forest | 14°30.158′ N | 101°55.481′ E | 449 |
| Doi Phu Kha N.P. | 1 | THAMH | Hill evergreen forest | 19°12.236′ N | 101°04.667′ E | 1341 |
| Doi Phu Kha N.P. | 2 | THAMI | Hill evergreen forest | 19°12.157′ N | 101°04.388′ E | 1327 |
| Doi Phu Kha N.P. | 3 | THAMJ | Hill evergreen forest | 19°12.311′ N | 101°04.846′ E | 1356 |
| Doi Phu Kha N.P. | 4 | THAMK | Hill evergreen forest | 19°10.447′ N | 101°06.368′ E | 1698 |

Collection dates and total numbers of parasitoid wasp specimens for each site are presented in Table 2.

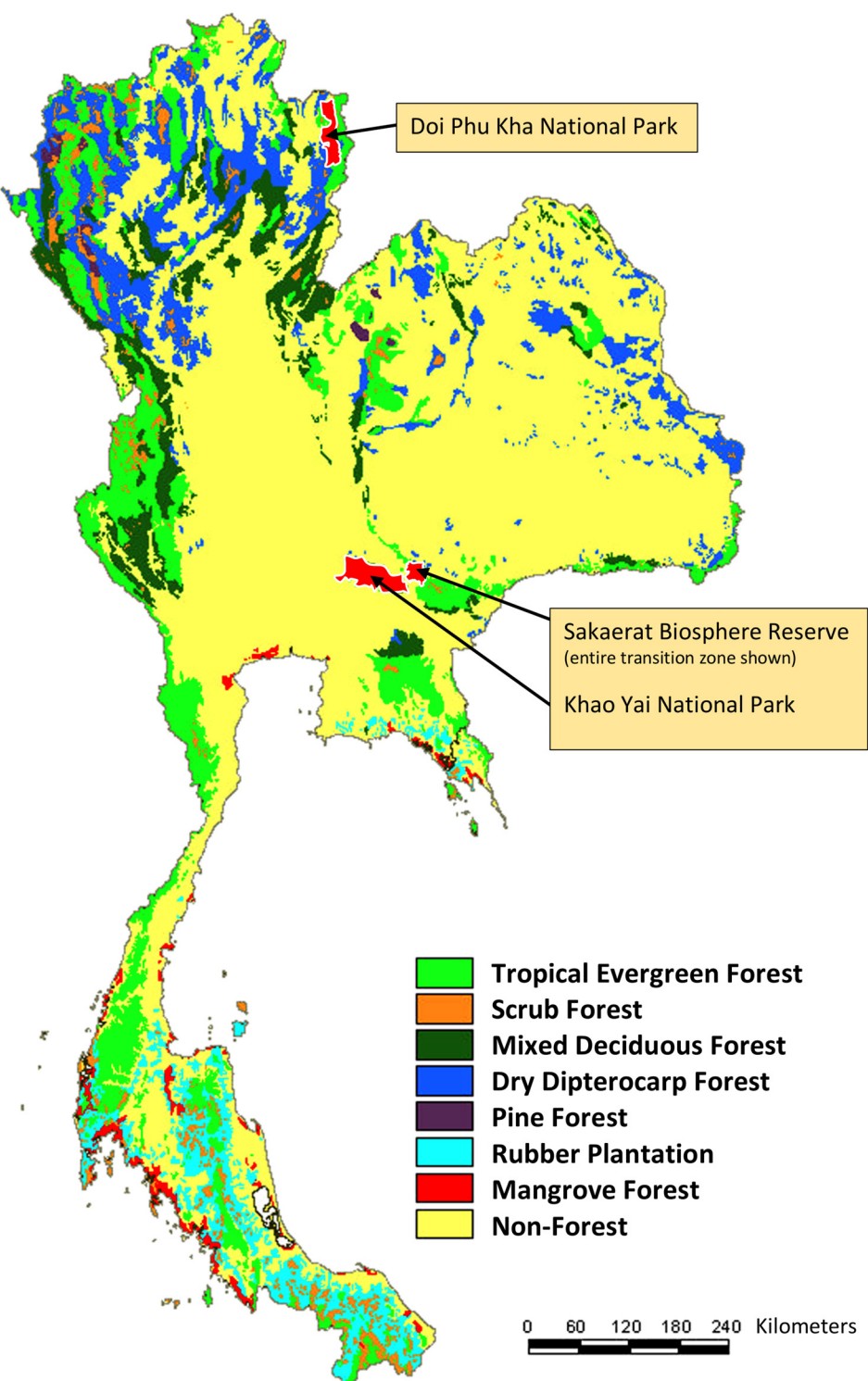

**Figure 1.** Map of Thailand showing locations of sampling areas, land use and vegetation type. (Base map produced by Roger Beaver and Sky Liu, reproduced with permission).

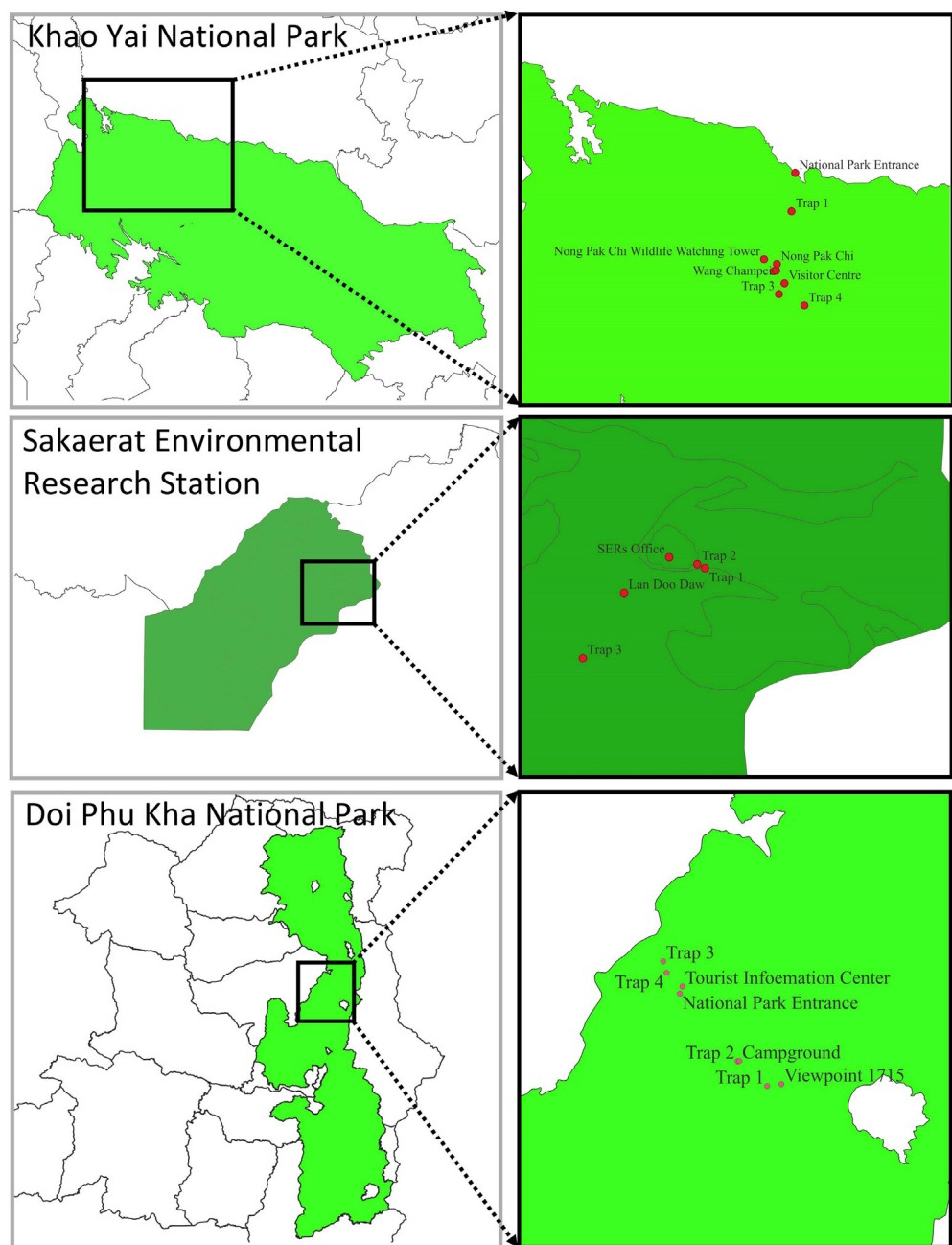

**Figure 2.** Expanded maps of the three sites showing precise locations of individual Malaise traps.

**Table 2.** Collection dates (week ending) for all trap samples and the number of hymenopteran parasitoids they contained. All analysed samples are from 2022. Hyphens indicate that the trap was not run.

| Location | Trap | 28.i | 25.ii | 25.iii | 22.iv | 20.v | 17.vi | 15–16.vii | 12–13.viii | 10.ix | 7.x |
|----------|------|------|-------|--------|-------|------|-------|-----------|------------|-------|-----|
| Khao Yai N.P. | 1 | 590 | 588 | 142 | 212 | 133 | 86 | 61 | 86 | - | - |
| Khao Yai N.P. | 2 | - | - | 232 | 179 | 176 | 151 | 129 | 99 | - | - |
| Khao Yai N.P. | 3 | - | 5 | 323 | 357 | 406 | 323 | 17 | 6 | - | - |
| Khao Yai N.P. | 4 | - | 9 | 869 | 701 | 528 | 446 | 8 | 158 | - | - |
| Sakaerat | 1 | - | 134 | 471 | 103 | 429 | 142 | 386 | 457 | - | - |
| Sakaerat | 2 | - | 91 | 264 | 251 | 362 | 194 | 462 | 219 | - | - |
| Sakaerat | 3 | - | 64 | 242 | 275 | 366 | 249 | 279 | 395 | - | - |
| Doi Phu Kha N.P. | 1 | - | - | - | - | - | - | 39 | 61 | 138 | 102 |
| Doi Phu Kha N.P. | 2 | - | - | - | - | - | - | 507 | 81 | 456 | 474 |
| Doi Phu Kha N.P. | 3 | - | - | - | - | - | - | 324 | 123 | 564 | 623 |
| Doi Phu Kha N.P. | 4 | - | - | - | - | - | - | 1373 | 63 | 48 | 68 |

*2.2. Specimen Photography and Barcoding*

Arthropod specimens were sorted by size for imaging and downstream molecular analysis. Small specimens (body length <5 mm) were placed individually into wells on a 96-well microplate pre-filled with ethanol, and DNA was non-destructively extracted from the whole specimen. Larger specimens were pinned, and a single leg was used for DNA extraction. Before DNA extraction, the small specimens were imaged with a Keyence VHX-7000 microscope using a custom script that takes a z-stacked image of each well. The larger specimens were imaged using a Canon SLR camera attached to a computer-controlled digital motor-drive rig, which allows automated capture of high-resolution photos of sets of 95 specimens arrayed on a pinning platform. All images were uploaded to BOLD (Barcode of Life Data System, http://boldsystems.org, accessed on 15 July 2023) along with full collecting metadata for each specimen.

To extract DNA from the legs and specimens, the ethanol was evaporated from the plates and replaced with 50 μL of tissue lysis buffer (1 M KCl, 20 mg/mL Proteinase K). The plates were incubated overnight at 56 °C. After lysis, DNA was purified using SPRI beads [18]. The COI barcode region was amplified through PCR using a cocktail of the Folmer primers [19] and LepF1 and LepR1 [20] and sequenced on the Sequel platform (Pacific Biosciences, San Diego, California, USA). For a detailed description of the PCR and sequencing protocols, see [21]. The resulting DNA barcode sequences were uploaded to BOLD for storage and analysis. The voucher specimens were deposited in the CBG voucher collection to preserve them for possible subsequent morphological analysis.

All barcoded specimens were identified to order level or below based on examination of both images and the barcode sequences. For the present study, a neighbor-joining tree of all Hymenoptera specimens was generated using the Kimura 2 parameter distance model after aligning the sequences with the amino-acid-based HMM aligner tool on BOLD. A matching specimen image library, where specimens are arranged in the order they appear in the tree, was generated as a part of the tree analysis on BOLD. Here, we deal only with the parasitoid Hymenoptera, of which 13,527 specimens were identified to the family level based on a review of the specimen images supplemented with information from where they appeared in the neighbor-joining tree. NCBI BLAST searches or the BOLD ID Engine were used to obtain further information on taxonomic placement for some specimens. Members of the Ichneumonoidea were further identified to subfamily level.

*2.3. Estimating Unknown Diversity*

All calculations were performed and all graphics were constructed using the statistical computing language R [22]. We used the R package SpadeR to calculate various estimators of the total number of BINs likely to be present at each site and across all sites based on our data [23,24]. The non-parametric estimator Chao 1 provides a conservative lower bound of the total number [25].

**3. Results**

*3.1. Total Parasitoid Hymenoptera BIN Representation by Family*

Of the 18,661 specimens of parasitoid wasps that were collected, 13,557 were confidently assigned to family, and their barcodes were assigned to BINs. In total, representatives of 46 parasitoid wasp families were included, which collectively belonged to 4917 BINs, but 95% of them belonged to only 16 families. The numbers of individuals of each parasitoid family collected in each trap are given in Table S1, and the numbers of BINs are represented in Table S2.

The samples sequenced thus far cover eight months at Khao Yai, seven at Sakaerat and four at Doi Phu Kha. However, the total number of weeks of trap catches represented a maximum of two months over that period. Combined barcode-based and morphological identification has revealed the first records for three families: Ismaridae (Diaprioidea), Diparidae and Signiphoridae (Chalcidoidea), in Thailand.

BIN Overlap between Sites and Traps

Although the habitat types of where the traps were set varied, those in Khao Yai and Sakaerat were all in a broadly similar and historically contiguous forested ecosystem. The traps in Doi Phu Kha were in a hilly evergreen forest located 500 km to the north. Therefore, we expected there to be a relatively greater overlap between the species collected in the first two sites compared with either Doi Phu Kha. This was indeed the case (Figure 3), with a maximum of just over 3% of BINs shared between Doi Phu Kha and the lower-elevation central Thai sites, compared with more than 11% between Khae Yai and Sakaerat. Only 47 BINs were shared by all three sites, twenty-one of which were Platygastroide, and ten were Ichneumonoidea. The figure indicates low overlap but considerably higher overlap (11%) between the two central Thailand sites with broadly similar vegetation and climate.

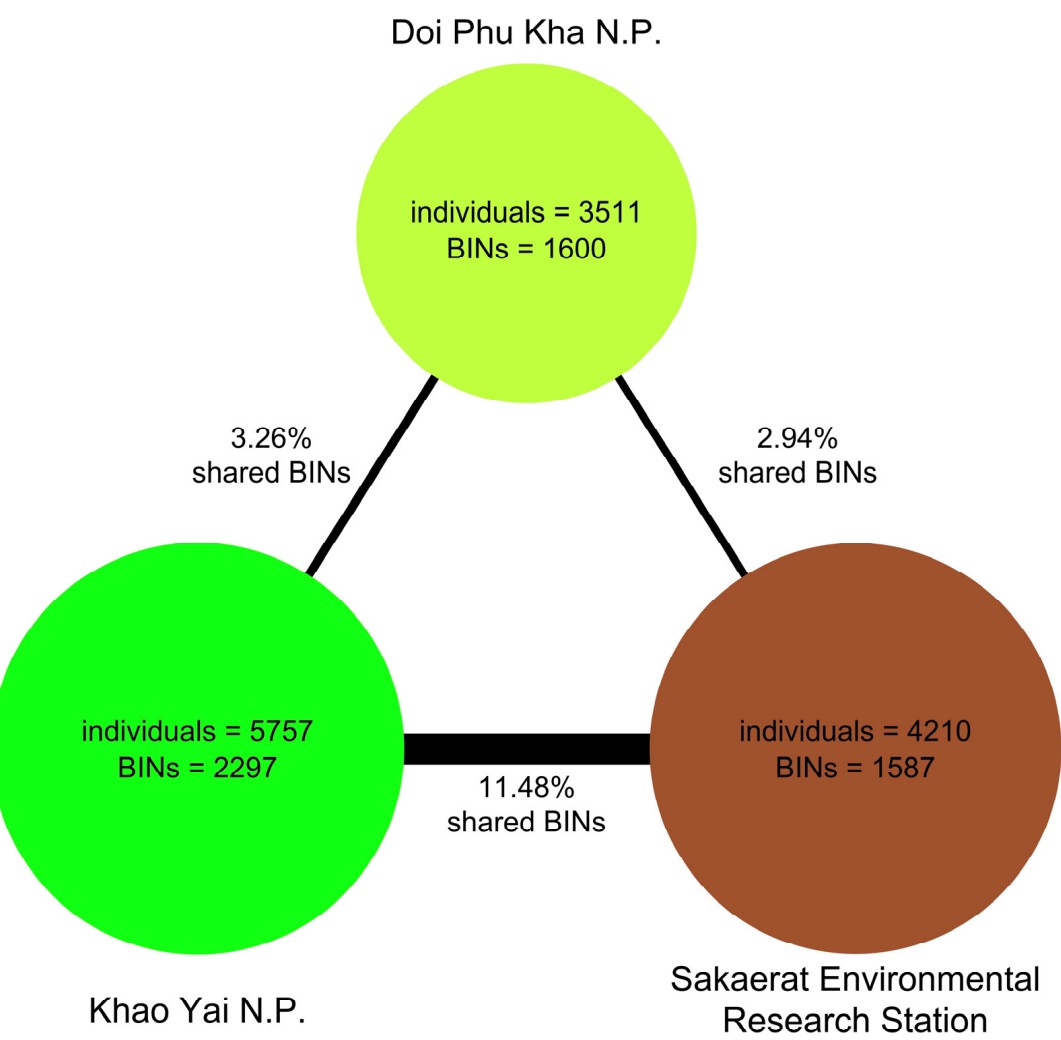

**Figure 3.** Chord plot showing percentages of parasitoid Hymenoptera BINs shared between each pair of localities.

*3.2. Family Representation*

By far, the most abundant family collected was Scelionidae, with just over 4000 successfully barcoded individuals (Figure 4), 4.2 times the number of unique BINs. In terms of life history and egg parasitoids (Scelionidae, Platygastridae, Mymaridae and Trichogrammatidae) were collectively represented by 1702 BINs and, thus, they constitute more than a third of the putative species collected.

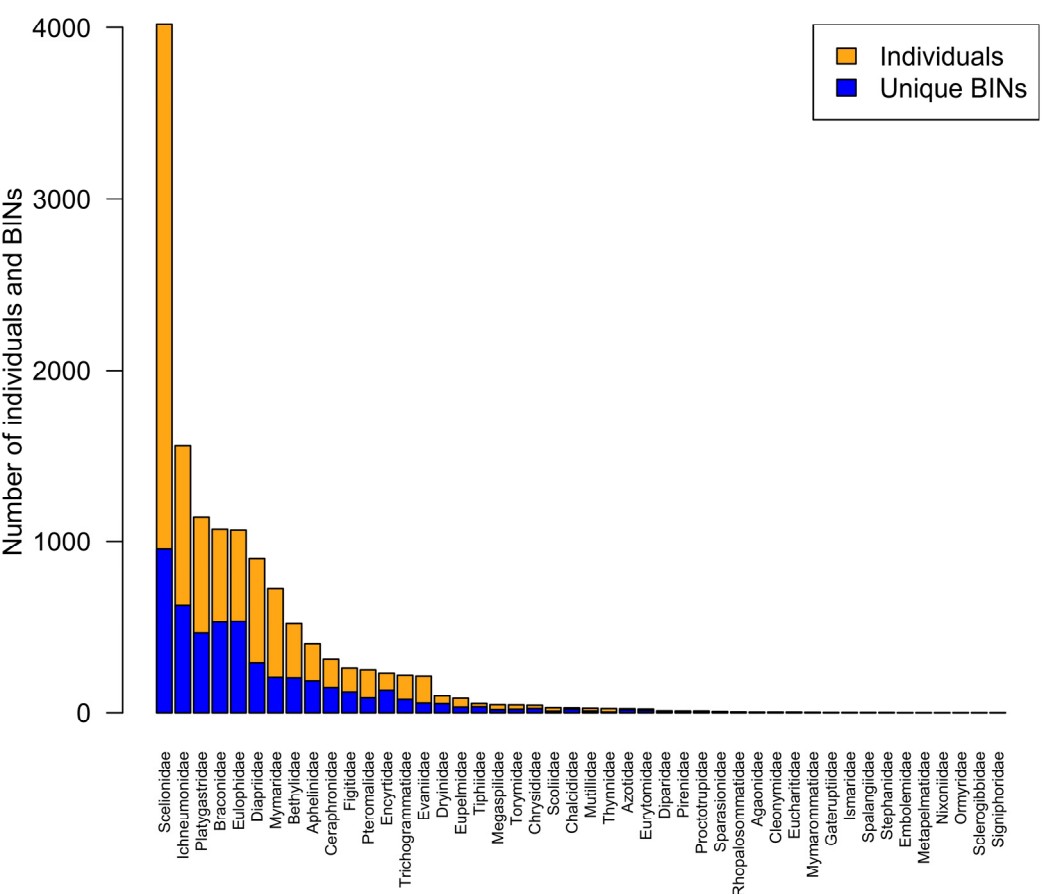

**Figure 4.** Representations of parasitoid wasps by family showing total number of individuals and number of unique BINs.

### 3.3. BIN Accumulation

The 66 weeks of samples from the 11 Malaise traps collectively contained representatives of 4917 parasitoid BINs. The smoothed species accumulation curve shows moderate evidence of a decreasing accumulation rate of new species with sampling effort (Figure 5), but it is virtually linear for the last four added traps.

There is a strong positive correlation between the number of parasitoid wasp specimens collected and the number of BINs discovered (Figure 6). On a linear scale, the relationship only for the most abundant family (Scelionidae) shows a slight tendency toward saturation.

### 3.4. BIN Extrapolation

All samples contained a high proportion of singleton BINs, and we employed the Chao 1 non-parametric equation to obtain a conservative estimate of the number of species of each family we might expect if sampling at the sites was continued for a much longer period. The observed numbers of BINs per family and the Chao1 estimated numbers are compared in Figure 7.

For the whole sample, 4917 BINs were observed. The conservative Chao1 prediction [25,26] was 9994 (S.E. = 269) with 95% confidence limits, 9493–10,549. The improved Chao1 (iChao1) [27] yields the best estimate of 10,864 (95% c.i. 10,470–11,285), and the abundance coverage estimator (ACE-1) [28] gives the largest estimate of 14,552 (95% c.i. 13,573–15,643).

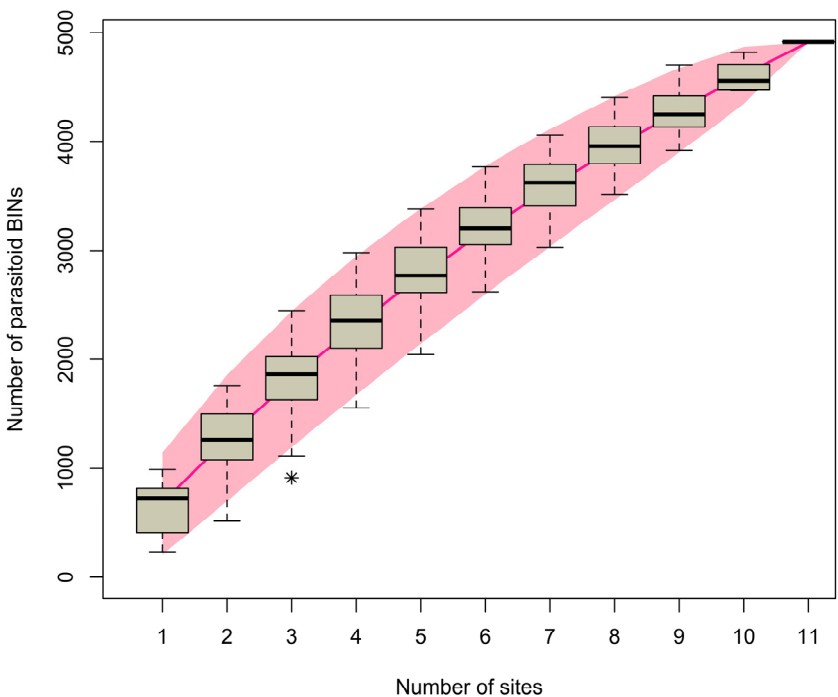

**Figure 5.** Total parasitoid BIN accumulation curve based on the combined catches from 11 traps produced using the function *specaccum* in the R library 'vegan'. Fitted line (red) is Lomolino model for exact accumulation [29]. The 95% confidence intervals are shown in pink. * $p < 0.05$.

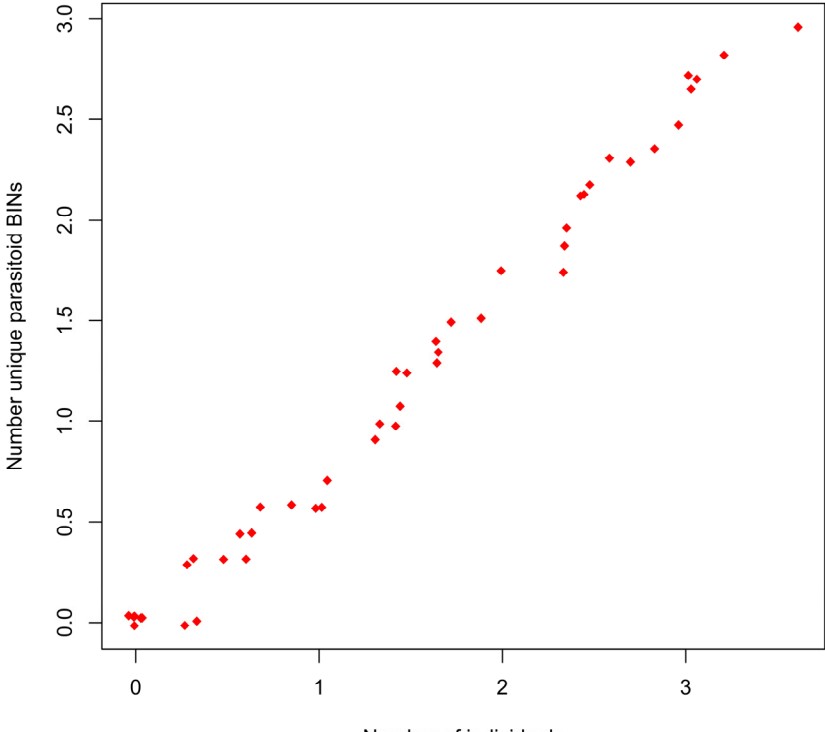

**Figure 6.** Relationship between $\text{Log}_{10}$(number of individual parasitoids collected) and $\text{Log}_{10}$(number of BINs detected).

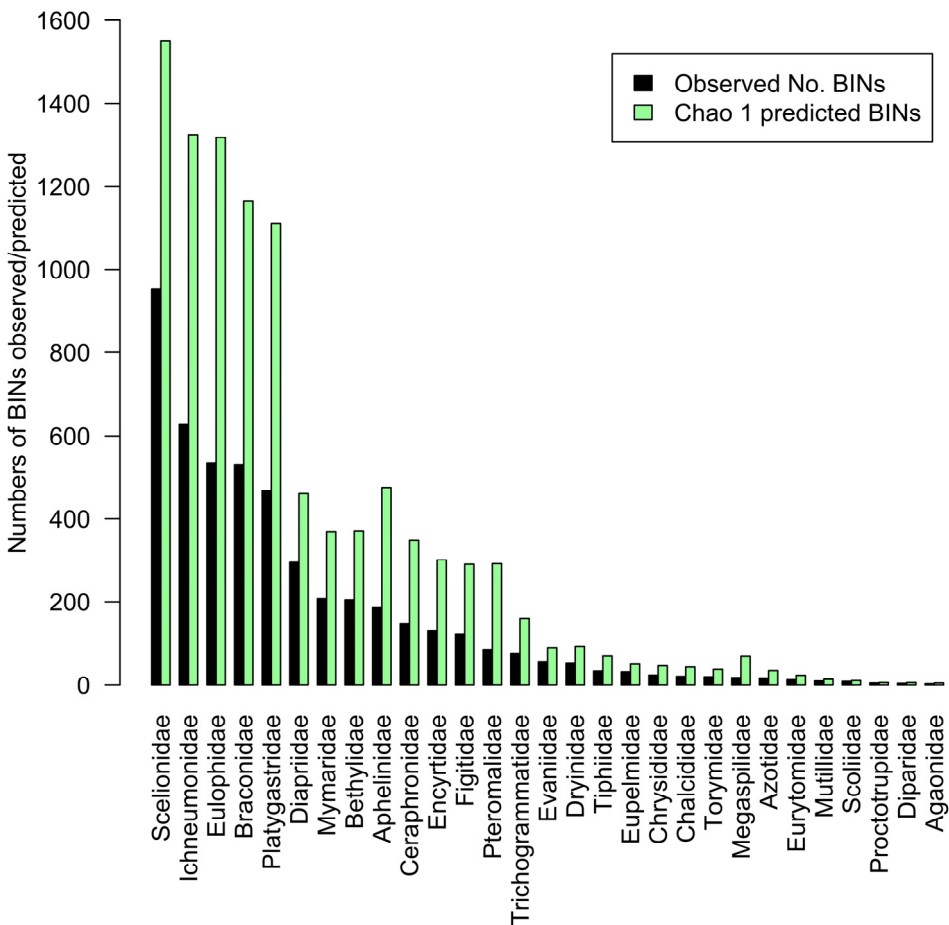

**Figure 7.** Observed numbers of parasitoid wasp BINs per family and Chao 1 non-parametric estimates of the total number at the sample sites.

### 3.5. Genus and Species Identification

Of the estimated 49,17 parasitoid species collected, 207 could be assigned confidently to genus, and only 55 had barcodes that belonged to BINs with a species designation (Table S3). See Section 4.1 below for a discussion of where these identifications come from.

### 3.6. Ichneumonoidea

Both families were very well represented, with Ichneumonidae being represented by 1552 BINs compared with 1065 braconids. Figures 8 and 9 show the subfamily distributions of BINs and compares them with the number of reported Thai species as of 2015, as listed on the Taxapad database [30].

It can be seen that for Ichneumonidae, the number of unique BINs from the three sampling sites exceeds the total number of species reported for the whole country and by a large factor in most cases.

The total number of Ichneumonidae individuals collected (Table S4) and numbers of BINs represented (Table S5) were greater than for Braconidae, and this was true of the majority of individual trap samples: eight out of eleven for individuals and nine out of eleven for BINs.

Figures 10 and 11 show the degree to which each trap shares BINs with each other trap for Braconidae and Ichneumonidae, respectively. The highest proportions of shared BINs, indicated by thicker chord lines, are between traps that are close to each other and in the central Thailand sites (Khao Yai and Sakaerat). However, the faunas of both families differed radically between the north Thai montane forest at Doi Phu Kha compared with the two central Thai localities, the same as for all parasitoids (see Figure 3).

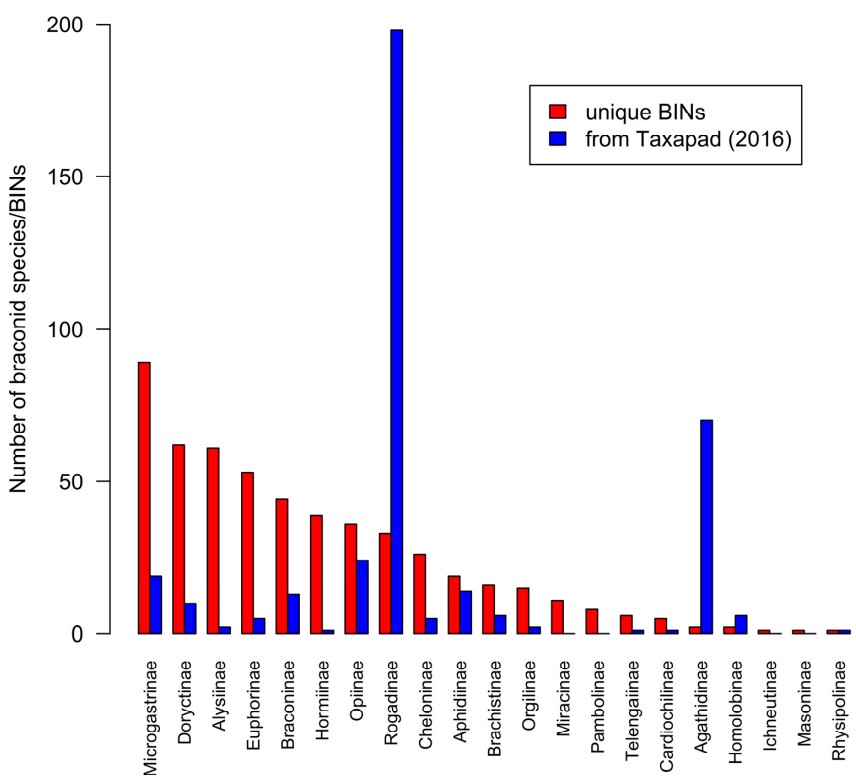

**Figure 8.** Ranked numbers of unique BINs per subfamily for Braconidae together with published numbers of Thai species for the whole of Thailand as of 2015.

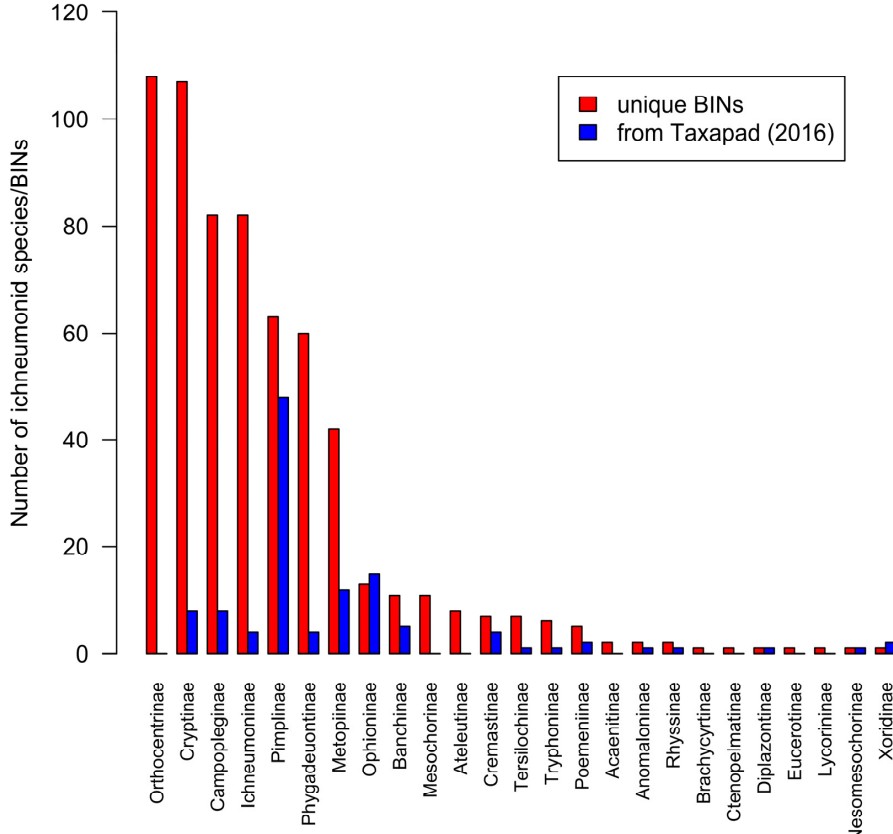

**Figure 9.** Ranked numbers of unique BINs for each of the 25 subfamilies of Ichneumonidae together with published numbers of species for the whole of Thailand as of 2015.

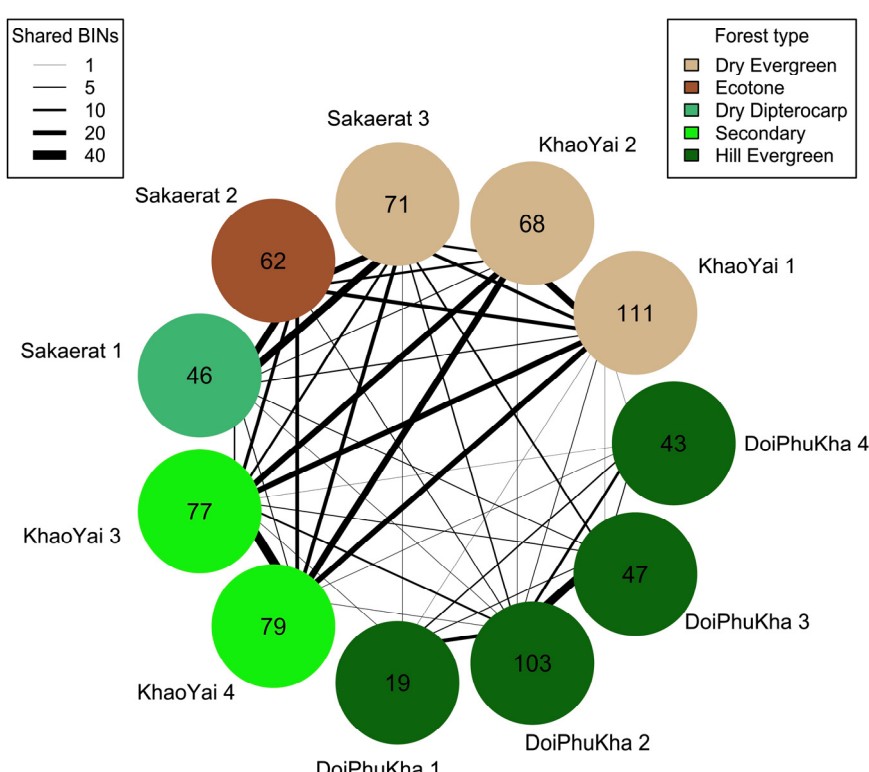

**Figure 10.** Chord plot showing percentages of Braconidae BINs shared between each pair of traps with forest types indicated by different colours.

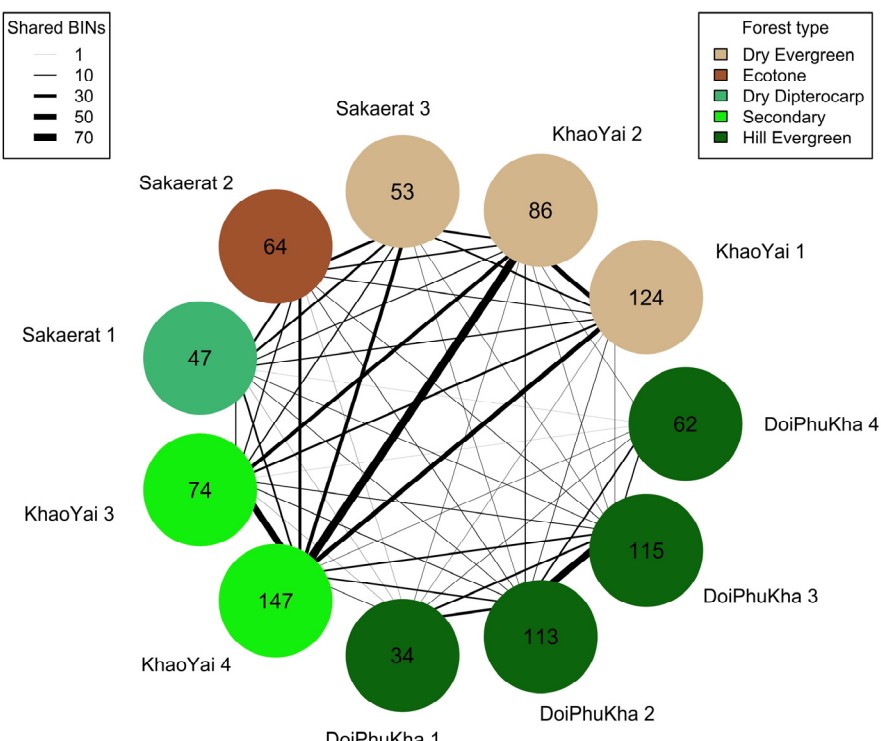

**Figure 11.** Chord plot showing percentages of Ichneumonidae BINs shared between each pair of traps with forest types indicated by different colours.

### 3.7. Life History Strategies

The life history strategy (idiobiont, koinobiont or imagobiont) is highly conserved at the subfamily level and is known for virtually all groups of Ichneumonoidea [10]. For both braconids and ichneumonids, koinobionts accounted for more BINs than idiobionts. Overall, koinobionts constituted 56.2% of BINs. In all cases, the proportions of koinobionts among BINs shared between sites (three pairwise comparisons) or of taxa shared across all three sites were greater. However, statistical tests ($\chi^2$ or Fischer's exact tests as appropriate) were not significant (all *p*-values > 0.2).

## 4. Discussion

Here, we used Barcode Index Number (BIN) [18] assignments as a proxy for species delimitation [31–33]. We recognise that some BINs include multiple biological species (e.g., when speciation has been rapid) or that members of a single species can be assigned to two or more BINs because of processes such as incomplete mitochondrial lineage sorting. However, we are confident that this system is considerably more accurate than any effort to separate the poorly studied Thai fauna into morphospecies. For extensive discussions of the use of BINs in the taxonomic separation of species, see [34–39]. In many less-well-studied groups, barcoding has been revealing the existence of surprisingly large numbers of undescribed species [40–42].

Malaise traps are well-known for not being very good at catching small-bodied Chalcidoidea, so chalcidologists normally concentrate their efforts on yellow pan trapping and screen sweeping. Nevertheless, a substantial number of small-bodied parasitoid wasps were represented in our samples (Chalcidoidea and Platygastroidea in particular). However, this collection bias suggests that other collecting methods would have substantially elevated the numbers of individuals and BINs in these groups.

### 4.1. Species Level Identifications

Out of the 4917 putative parasitoid species collected, 207 could be assigned to genus, and 55 could be named to species. The sources of species-level identifications were varied. Twelve species were identified because BOLD contained sequences representing named specimens sequenced in other projects, eleven were identified based on sequences that had been submitted to GenBank for currently unpublished research, ten were based on sequences submitted to GenBank as a part of published molecular phylogenetic studies [43–48], eight were from taxonomic revisions of various genera [13,49,50] that included barcode data, four were from studies of particular pest species and their parasitoid complexes [51–54], three were from a barcoding study of Canadian insects [33,55], two were from a barcoding inventory of Finland [56], and one was from the barcoding of German specimens held in a German museum [57]. The remainder are widespread, often commercially important parasitoids of crop or forestry pests; for example, the aphid parasitoids, *Binodoxys acalephae* and *Lipolexis oregmae* (both Braconidae, Aphidiinae) are important and widespread parasitoids of *Aphis gossypii* and numerous other aphids, and the almost-cosmopolitan polyembryonic encyrtid, *Copidosoma floridanum*, is a major and important parasitoid of many pest species of plusiine noctuid moths and, no doubt, of many more non-pest species [58,59]. Only in the case of the five species of the rogadine braconid genus *Aleiodes* were the identified species from part of a Thailand-based taxonomic study [13].

The only other parasitoid group specifically revised for Thailand is the braconid subfamily Agathidinae [60], with several subsequent genus revisions [61–63]. However, although the described Thai fauna comprises 70 species, neither of the two specimens (two separate species) in our samples were among the many that have been barcoded.

Many of the identified species have wide geographic distributions and are associated with pest insects as parasitoids or hyperparasitoids. For example, *Coccophagus bogoriensis* (Aphelinidae), which parasitises various scale insects (Coccidae and Diaspididae), was originally described in Indonesia but is now distributed throughout China and India and has been introduced in the West Indies and Neotropical region. Several are species

originally described from Europe, e.g., the microgastrine braconid, *Diolcogaster connexa*, and the scelionid, *Telenomus turesis*. Most of the other identified species are distributed in the east Palaearctic, e.g., the opiine braconid *Coleopioides postpectalis* is known to be present in China and South Korea.

In terms of barcode coverage of named insect species, the data for Germany and Fennoscandinia are best [56,64–66]. For Germany, Geiger et al. [67] presented data on the proportions of all insect BINs identified in Global Malaise Trap Program samples (based on 37,274 specimens classified into 5301 BINs) that could be identified to various levels through reference to existing barcode databases. In total, 35% could be unambiguously identified at the species level and a further 7% at the genus level. In comparison, just dealing with parasitoid Hymenoptera, only 4.2% of species could be identified at the genus level, and just over 1% at the species level. We do not know what proportion of these identifications were made possible through species having been included in molecular phylogenetic studies or having been investigated from a biocontrol perspective. It is likely that both are somewhat higher than for the Thai samples, but likely not by much. Probably, the great majority are the result of barcode data having been deliberately collected from named specimens of European insects. This huge disparity highlights the need for developing barcode databases for accurately identified tropical insects because without such databases, barcoding is of little value in assigning Linnean names if this is the goal of a study. However, as this study shows, much can be learned about the composition and regional diversity patterning of a fauna without their application.

### 4.2. Anomalous Diversity of Tropical Ichneumonidae

It has long been recognised that some groups of Ichneumonidae (and a few Braconidae) have far more temperate-region-centred distributions than others because their hosts are poorly represented in the tropics. These include members of the ichneumonid subfamilies Ctenopelmatinae and Tryphoninae (most tribes), whose hosts are sawflies, and aphidiine braconids, whose hosts are aphids [10]. However, there was, for a long time, no reason to suspect that subfamilies attacking tropico-centric hosts, such as Lepidoptera and Coleoptera, would not also display their highest diversities in the tropics.

Following comparisons of the Malaise trap and sweeping samples between temperate and tropical sites, refs. [68–70] an unexpected finding emerged, namely that the tropical samples consistently contained fewer species of Ichneumonidae than their temperate equivalents. This became known as anomalous diversity, and so, Ichneumonidae joined various other groups with relatively low tropical diversity, such as bumblebees (Apidae: Bombinae), penguins and freshwater zooplankton. Based on the assumption that Ichneumonidae displayed anomalous diversity, several explanatory hypotheses were developed, for example, one based on higher tropical predation pressure [71], another, the 'nasty host hypothesis' [72], based on some evidence that host foodplants in the tropics generally contained higher levels of secondary plant compounds than their temperate equivalents, and this resulted in more potential hosts occupying enemy-free spaces. Sime and Brower [73] reviewed the evidence for and against each proposal and concluded that the 'nasty host hypothesis' was the most credible [10].

Conclusions about insect tropical diversity have generally been based on the number of species described or recorded from each region. For well-studied groups, such as butterflies and hawkmoths, this probably provides fairly good estimates. However, parasitoid wasps have received far less taxonomic attention, so the rate of the description of their species, especially of small, less charismatic ones, lags quite far behind. Quicke [74] analysed the available species records of both Braconidae and Ichneumonidae in relation to latitude and compared the distributions with those of the much more completely studied mammals and angiosperm plants.

Recent studies in the tropics have started to reveal more species of Ichneumonidae than previously suspected [75–77]. In particular, there appeared to be proportionately far more undescribed species of the taxonomically neglected subfamily Orthocentrinae.

These small-bodied insects are parasitoids of Diptera larvae occupying decaying substrates, particularly fungus gnats (Mycetophilidae), although there are extremely few definite host records [78]. Our data highlight the understudied nature of this group (Figure 9). We detected 107 different Orthocentrinae BINs, but not a single species had been reported from Thailand as of 2015 [30]

As of 2019 [79], the known Thai Ichneumonoidea fauna included 97 species of Ichneumonidae and 377 species of Braconidae. The number of described species only exceeds BINs for Ophioninae and Xoridinae (Figure 9). The former are generally large-bodied and are nearly all readily attracted to lights at night and so have been revised for the Indopapuan region [80]. Xoridinae are also large-bodied and, as a subfamily, very easy to recognize; therefore, it is not surprising that several species have been recorded. Pimplinae are also fairly well represented among Thai records, and this group has been generally receiving quite a lot of taxonomic attention in the region.

Regarding Braconidae, the overall picture is similar to that for Ichneumonidae, with the exception of two subfamilies, Agathidinae and Rogadinae (Figure 8). The first of these was specifically revised for Thailand by Sharkey and Clutts [60] based on an enormous amount of material collected by Mike Sharkey's TIGER (Thailand Insect Group for Entomological Research) project, a USA National Science Foundation (NSF) funded biodiversity inventory survey of the country, which utilised 3605 Malaise traps serviced at 7 day intervals at 559 separate sites spread across more than 30 Natural Parks and reserves between 2006 and 2009 and that ran for a year with the assistance of numerous local assistants [60,81]. The large number of Rogadinae reflects a revision of the Thai species of the single (but large) genus *Aleiodes* [13], which was largely based on the same TIGER project samples and included as much barcode data as possible. That study increased the number of described Thai *Aleiodes* species from 7 to 186, with descriptions of 179 new species. Further, applying the Chao 1 estimator [24] at that time led the authors to propose that the actual number of Thai species was probably at least 478.

### 4.3. Total Thai and Global Parasitoid Wasp Diversity

In common with several other studies e.g., [82,83], the overall species accumulation curve (Figure 5), as well as those for each individual family, showed no indication of plateauing, and therefore, even at the current three localities, we have thus far not sampled nearly enough to compile near-complete inventories.

It should be noted that many habitat types (the majority) in Thailand have not been sampled at all in the present study. All the traps deployed so far are in the central northern part of the country. The Isthmus of Kra in the south of Thailand marks the boundary between two very distinct faunas, the Indochinese to the north and the Sundaic zoogeographic regions [84–88]. In addition, the only real tropical rain forest in Thailand occurs in the extreme south (protected in Khao Sok National Park), south of the Isthmus of Kra. These two factors combined probably indicate that once the whole country is surveyed, a very large increase in the numbers of species is likely.

As of the beginning of 2022, 11,486,730 insect barcodes were available [89]. Despite that, strictly tropical insects, in general, are vastly understudied, with very few barcodes available for the identified species in most groups. This is exemplified in our study, with DNA-based species identifications only available for 55 BINs (see Section 4.2). All of these, except for the five members of the braconid genus *Aleiodes*, which has been revised for Thailand, are species that are considerably more widespread, some are cosmopolitan and mostly associated with pest species. Putting names to even just the majority of the remainder is likely going to require full revisions of all the other represented genera in the region. Further, such revisions will need to provide barcodes for as many species as possible. Ideally, the museums holding most of the historical-type material of tropical insects should seriously consider allowing for the use of modern DNA technology to try to retrieve barcode data from their precious specimens.

**Supplementary Materials:** The following supporting information can be downloaded at: https://www.mdpi.com/article/10.3390/f14101991/s1. Table S1: Frequencies of the individuals of each parasitoid hymenopteran family in samples from each trap locality; Table S2: Numbers of unique parasitoid BINs per Malaise trap per family. Individual Malaise traps designated by the letters A to K correspond to Khao Yai traps 1–4, Sakaerat traps 1–3 and Doi Phu Kha traps 1–4, respectively; Table S3: species-level identifications based on BIN membership; Table S4: number of individual Ichneumonidae Braconidae specimens per trap. The larger value for each trap is in bold font; Table S5: number of Ichneumonidae Braconidae BINs per trap. The larger value for each trap is in bold font. Figure S1A–K: photographs of the 11 Malaise traps in situ: A–D, Traps 1–4 in Khao Yai National Park, respectively; E–G, Traps 1–3 at Sakaerat Environmental Research Station, respectively; H–K, Traps 1–4 in Doi Phu Kha National Park, respectively.

**Author Contributions:** Conceptualization, D.L.J.Q., B.A.B. and P.D.N.H.; methodology, D.L.J.Q.; software, D.L.J.Q.; validation, D.L.J.Q., B.A.B. and M.P; formal analysis, D.L.J.Q.; investigation, M.P. and D.L.J.Q.; resources, P.D.N.H. and B.A.B.; data curation, M.P.; writing, original draft preparation, D.L.J.Q.; writing, review and editing, D.L.J.Q., B.A.B., M.P. and P.D.N.H.; visualization, D.L.J.Q. and P.D.N.H.; supervision, B.A.B.; project administration, B.A.B.; funding acquisition, P.D.N.H. and B.A.B. All authors have read and agreed to the published version of the manuscript.

**Funding:** This research was funded by the National Research Council of Thailand (NRCT) (N42A650262) and Chulalongkorn University, RSPG-Chula to BAB D.L.J.Q. was supported by the Rachadaphisek Somphot Fund for postdoctoral fellowship, Graduate School, Chulalongkorn University. Sequence analysis was supported by a Transformation 2020 award to P.D.N.H. from the New Frontiers in Research Fund, while critical infrastructure at the CBG was acquired with grants from the Gordon and Betty Moore Foundation and the Canada Foundation for Innovation (CFI). A Major Science Infrastructure award from CFI sustained the CBG's capacity to provide informatics and sequencing support.

**Data Availability Statement:** The DNA barcode sequences of the parasitoid wasps analysed, along with the specimen images and full collecting data, are available as a public dataset on BOLD (DS-pending, doi: pending).

**Acknowledgments:** We are very grateful to the following for help in arranging Malaise trap deployments and organising sample collections: Phasin Inkeaw (Doi Phu Kha National Park), Surachit Wangsothorn (Director of Sakaerat Environmental Research Station), Kanoktip Somsiri (Expert Centre of Innovative Clean Energy, and Environment, Thailand Institute of Scientific and Technological Research (TISTR)) and Weeraphol Suebjaksri (Khao Yai National Park). Kittipum Chansri kindly produced the maps showing individual trap locations.

**Conflicts of Interest:** The authors declare no conflict of interest.

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
