# Peer review of "Barcoding Hymenoptera: 11 Malaise Traps in Three Thai Forests: The First 68 Trap Weeks and 15,338 Parasitoid Wasp Sequences"

_forests, doi:10.3390/f14101991_

Round 1
Reviewer 1 Report
this is a very interesting paper that reports on a large gap in our knowledge of species diversity in the tropics. it will be of interest to a wide range of readers.
lines 72-73 out of place?, could be moved to lines 51?
Fig1 there is a problem with the legend, looks like the legend is pasted >1 and so is blurry
Fig 1 can you place arrows from the names of the Parks to their location on the map
Table 2. what does "0" mean in the table - either 1) nothing was caught OR 2) no sampling was completed. Please explain
line 206 this is a mistake, it should read 4,917 NOT 49,173!
figures - some of the text on the X axis is hard to clear - but perhaps these are low resolution images
one paper not in ref but of relevance is [Saunders TE & Ward DF. 2018. Variation in the diversity and richness of parasitoid wasps based on sampling effort. PeerJ 6:e4642 https://doi.org/10.7717/peerj.4642] showing even with high numbers of Malaise traps, saturation of ichneumonidae species was not reached
Author Response
Referee 1.
Lines 72-73. We have moved these as suggested.
Fig. 1. Nothing is blurry in the original, the blurriness is a consequence of a pdf being within another pdf. We will ensure it is clear in the final published version. We have encountered this issue previously with our MSS submitted to MDPI journals.
Fig. 1. RE: the request to place arrows!! There ARE arrows on our figure, again we think this is another issue pdf being within another pdf problem. We will resolve it.
Table. 2. Yes, 0 meant the trap was not run in that time period. We have replaced zeros woth hyphens and added a comment in the table title.
Line 206. We have corrected this typo.
Figure legends hard to read. Nothing is hard to read in the originals, this is a consequence of a pdf being within another pdf. We will resolve it.
We thank the referee for drawing our attention to the reference by Saunders and Ward, which we found interesting and have now referred to in the MS at the beginning of Section 4.3. So that it fits better, we also refer to another additional paper of parasitoid wasp accumulation curves.
Reviewer 2 Report
This study makes a valuable contribution to our understanding of tropical hymenopterans using a combination of Malaise traps and molecular techniques. The authors have employed Barcode Index Number (BIN) assignments as a proxy for species delimitation, and they have provided a thoughtful discussion of the merits, drawbacks, and contributions of this approach to the Thai fauna. While the paper holds promise for publication, there are areas that require revision, particularly in the mixed expression of methodology, results, and discussion sections. Below are specific points that need attention:
- Line 76: In the review document, the resolution of the pictures is insufficient. It appears that the Bugdorm Malaise traps used may involve different generation products. While this might not significantly affect the results, it should be indicated in the Material and Methods section if different trap generations were employed.
- Line 80: The statement "samples were collected weekly" lacks clarity regarding the collection interval and duration. Please rephrase to provide a clearer explanation.
- Line 84: The resolution of the map is inadequate, making it difficult to discern the boundaries of the three conservation areas. Additionally, it would be helpful to label the sampling sites of the Malaise traps on the map instead of just displaying vegetation types.
- Line 137: The statement regarding the duration and sampling frequency, "the samples sequenced thus far cover just three months at each site and represent only three sampling weeks during that period," appears in the Results section (3.1). This information should be clearly stated in the Materials and Methods section.
- Line 141: The results presented in section 3.1.1 concerning "BIN Overlap Between Sites and Traps" are intriguing. However, the results do not seem to align with habitat types as expected. This discrepancy warrants a more in-depth discussion. Furthermore, it appears that the influence of temporal factors versus habitat types may have been overlooked. For instance, consider the case of traps THAMA and THAMB, which share similar habitats but exhibit significantly different numbers of captured hymenopterans in 28.i and 25.ii (Table 2). This observation underscores the need for a comprehensive exploration of these factors.
- Lines 243-250: Section 3.5.1 describes the results of life history strategies but combines elements of Materials and Methods, results, and discussion in a mixed manner. It would be beneficial to present this information more clearly and separately in each relevant section.
Lines 628 and throughout the manuscript: The notation "Individual Malaise traps are designated by the letters; A to K correspond to Khao Yai traps 1-4, Sakaerat traps 1-3, and Doi Phu Kha traps 1-4, respectively" provides a link between the letter and number assignment to each trap which appears in multiple places in either way. Consider integrating this information into a table or providing it once in a consolidated format, especially since detailed trap information is already available in Table 1. This will streamline the presentation and avoid confusion in the manuscript.
Author Response
Referee 2.
- Line 76. We have provided higher resolution images of each of the traps in situ as supplementary files S1-S11. We assure the referee that all traps are of the same model, so there are no issues of bias.
- Line 80. We thought that 'weekly' was explicit enough, but we he have changed it to "every 7th day" to be totally precise.
- Line 84. The purpose of this map was to show readers unfamiliar with the geography of Thailand where the sampling sites are and their relative separation. We have added a new Figure 2 showing the precise locations of the traps at each of the three sites. In Supplementary Figures S1-S11 we provide maps showing accurately where in each of the protected areas, the individual Malaise traps were located.
- Line 137. We have corrected and clarified our error (left over from an earlier draft) about the duration of the trapping period.
- Line 141. We thought we had commented on this aspect sufficiently given that we do not have many independent pairs of traps in separate blocks of the same forest classification, and feared that we might have been criticised had we done so. In light of the referee's comments, we have written a little bit more about it, and stressed that it needs more work. Indeed, we have traps in similar habitat at a more remote site but those samples will not be processed for some while.
- Lines 243-250. We have followed the referee's advice and separated the elements into the relevant sections.
7. We used A-K and, e.g. "Khao Yai trap 1" because of our desire to present tables neatly where the longer form would prevent them fitting within the available width. We have changed to the long version in Table 2, but have left the letter codes in Table S1 because of the spacing issue, but defined their meanings in the legend.